# Targeting Anterior Commissure Involvement with Hyperfractionated Radiotherapy for T1–T2 Squamous Cell Carcinoma of the Glottic Larynx

**DOI:** 10.3390/cancers16101850

**Published:** 2024-05-12

**Authors:** Satoshi Seno, Kazuma Iwashita, Akifumi Kajiwara, Rie Sasaki, Tatsuya Furukawa, Masanori Teshima, Hirotaka Shinomiya, Naomi Kiyota, Rod Lynch, Kenji Yoshida, Takeaki Ishihara, Daisuke Miyawaki, Ken-ichi Nibu, Ryohei Sasaki

**Affiliations:** 1Division of Radiation Oncology, Kobe University Graduate School of Medicine, Kobe 650-0017, Japan; sase@med.kobe-u.ac.jp (S.S.); kziwst89@med.kobe-u.ac.jp (K.I.); kajihara@med.kobe-u.ac.jp (A.K.); riaakua@med.kobe-u.ac.jp (R.S.); take3036@med.kobe-u.ac.jp (T.I.); miyawaki@med.kobe-u.ac.jp (D.M.); 2Department of Otolaryngology-Head and Neck Surgery, Kobe University Graduate School of Medicine, Kobe 650-0017, Japan; ftatsuya@med.kobe-u.ac.jp (T.F.); mateshim@med.kobe-u.ac.jp (M.T.); hshino@med.kobe-u.ac.jp (H.S.); nibu@med.kobe-u.ac.jp (K.-i.N.); 3Kobe University Hospital Cancer Center, Kobe 650-0017, Japan; nkiyota@med.kobe-u.ac.jp; 4Department of Radiation Oncology, Andrew Love Cancer Centre, Barwon Health, Geelong, VIC 3220, Australia; rod.lynch@barwon-health.org.au; 5Division of Radiation Oncology, Tottori University, Yonago 680-0945, Japan; kyoshi@tottori-u.ac.jp

**Keywords:** anterior commissure involvement, glottic cancer, hyperfractionation, laryngeal preservation

## Abstract

**Simple Summary:**

Anterior commissure involvement (ACI) is an important factor in the local control of early-stage glottic squamous cell carcinoma (EGSCC). We initiated a radiotherapeutic treatment regimen focusing on ACI, which has included hyperfractionated radiotherapy since 2008. One-hundred and fifty-three patients with T1–T2 EGSCC were included in this study. Hyperfractionated radiotherapy (74.4 Gy in 62 fractions) was used for EGSCC with ACI. The 10-year overall survival and cause-specific survival rates were 72% and 97%, respectively. The 10-year local control rates were 94% for T1a, 88% for T1b, and 81% for T2 disease. Local control rates in patients with ACI were slightly better than those in patients without ACI with T1a and T1b diseases; however, the difference was not significant. The 10-year laryngeal preservation rate was 96%. In conclusion, hyperfractionated radiotherapy was effective for T1 disease with ACI but insufficient for T2 disease with ACI. Our treatment strategy resulted in excellent laryngeal preservation.

**Abstract:**

Anterior commissure is involved in about 20% of early-stage glottic squamous cell carcinomas (EGSCCs). Treatment outcomes and prognostic factors for EGSCC with anterior commissure involvement (ACI) were evaluated by focusing on hyperfractionated radiotherapy (74.4 Gy in 62 fractions). One-hundred and fifty-three patients with T1–T2 EGSCC were included in this study. The median total doses for T1a, T1b, and T2 were 66, 74.4, and 74.4 Gy, respectively. Overall, 49 (32%) patients had T1a, 38 (25%) had T1b, and 66 (43%) had T2 disease. The median treatment duration was 46 days. The median follow-up duration was 5.1 years. The 10-year overall and cause-specific survival rates were 72% and 97%, respectively. The 10-year local control rates were 94% for T1a, 88% for T1b, and 81% for T2 disease. Local control rates in patients with ACI were slightly better than those in patients without ACI with T1a and T1b diseases; however, the difference was not significant. The 10-year laryngeal preservation rate was 96%. Six patients experienced grade 3 mucositis, and four patients had grade 3 dermatitis. Hyperfractionated radiotherapy was effective for T1 disease with ACI, but insufficient for T2 disease with ACI. Our treatment strategy resulted in excellent laryngeal preservation.

## 1. Introduction

Laryngeal squamous cell carcinoma comprises about 5% of all head and neck cancers [1]. Glottic squamous cell carcinoma represents approximately 75% of laryngeal malignancies [2], and over 80% of these patients present with an early Union for International Cancer Control (UICC) stage [3]. The larynx plays an important role in phonation, coordination of swallowing, and respiration. Because the larynx has important functions, the treatment goal for laryngeal cancer is not only to control the disease but also to preserve the functional larynx.

In the absence of large randomized clinical studies demonstrating obvious evidence for the ideal strategy for treating early-stage glottic squamous cell carcinoma (EGSCC), lots of retrospective studies have reported comparable control rates after surgery or radiotherapy. The 5-year local control (LC) rate following radiotherapy is 80–95% for T1 and 61–82% for T2 disease. Five-year overall survival (OS) rates for stage I and II disease are 89–100% and 60–100%, respectively [4,5]. Both transoral laser microsurgery (TLM) and radiotherapy provide favorable voice quality in T1 and T2 disease, whereas open vertical partial laryngectomy results in poor voice quality [6,7,8].

The anterior commissure (AC) is rarely the original site of these tumors, but it is involved in up to 20% of EGSCC cases [9,10]. The AC forms a subsite of the larynx that exhibits vertical extension according to its embryonic origin [11]. Owing to the absence of the inner perichondrium, which is connected to the intermediate lamina of the thyroid cartilage [12,13], the AC is particularly vulnerable. Specifically, the disease erodes through a part of the cartilage; therefore, the AC has always been a subject of anatomical, diagnostic, and therapeutic controversy in laryngeal oncology.

Tumor-related prognostic factors for EGSCC are stage [14], tumor extension [15], AC involvement (ACI) [16,17,18,19], anemia [20], and continuation of smoking [21], on which the impact of dose/fractionation has been extensively investigated. Multiple randomized studies have demonstrated the benefit of shorter treatment times, regardless of whether the goal was achieved using accelerated fractionation or hypofractionation [22,23,24,25,26].

Furthermore, the pooled difference in the 5-year local recurrence rates between T1 EGSCC without (T1a) and with (T1b) ACI is reportedly as high as 12%, regardless of the type of treatment delivered (either radiotherapy or TLM). Although several studies have shown a significant association between ACI and a higher recurrence rate of EGSCC, the specific value of the AC is not considered in the current TNM staging system [27].

With this in mind, we implemented an original aggressive treatment regimen using hyperfractionated radiotherapy for treating tumors with ACI. In this study, we evaluated the treatment outcomes of EGSCC, specifically focusing on the significance of ACI. The aim of this study was to clarify whether adaptation of hyperfractionated radiotherapy of EGSCC with ACI may contribute to local controllability and ultimately excellent laryngeal preservation.

## 2. Materials and Methods

### 2.1. Patients and Treatment

A total of 161 consecutive patients with histologically proven EGSCC (T1–T2N0M0, UICC, 6th edition) were investigated. The patients were treated with radiotherapy between January 2008 and November 2021. Patients treated with TLM were not included. One patient treated with altered fractions and seven patients treated with chemoradiotherapy were excluded from this study. Consequently, 153 patients were included in this study. The average number of cases per year was 11 (range: 3–23). For these patients, the period of follow-up management was for 15 years, between June 2008 and August 2023. Detailed information about the patients and the characteristics of the tumors are presented in Table 1. All patients provided written informed consent for the study and agreed to use accompanying images. This retrospective analysis was approved by the Institutional Review Board of Kobe University Hospital (No. B230238). The decision-making process for determining the initial therapeutic strategy was employed as described previously [28]. Briefly, each new case was presented and discussed at the Head and Neck Cancer Board Conference of the Kobe University Hospital. The board conference consisted of each specialist: head and neck surgeons, radiation oncologists, medical oncologists, and radiologists. Staging and treatment choice were debated at this board conference to determine the best treatment strategy for preserving a functional larynx. The staging workup included a physical examination, a blood test, video laryngoscopy, and multidetector computed tomography (CT) of the neck and thorax (thin slice [1 mm thickness] images). Close assessments were performed with respect to performance status, tumor location in the larynx, tumor volume, patient age, presence of other cancers, and other conditions (including intercurrent illnesses). Possible salvage treatments in the event of local failure were initially discussed at the head and neck cancer board conferences. During the conference, total laryngectomy was avoided to preserve vocal function. We classified AC tumors into the following three categories according to the ACI size: AC0, without ACI; AC1, with ACI but without a bulky tumor; and AC2, with ACI and a bulky tumor (Figure 1). In accordance with Reddy et al. [29], we observed defined bulky tumors through laryngoscopy and defined them as follows: large and/or infiltrative neoplasms involving an entirely true vocal cord and horseshoe-shaped lesions involving more than the anterior one-third of both true vocal cords.

### 2.2. Radiotherapy

All patients were treated with external-beam radiotherapy using high-energy photons from a 4-MV X-ray linear accelerator (Varian Medical Systems Inc., TrueBeam, Tokyo, Japan). For treatment planning, patients were immobilized using a thermoplastic mask, before the acquisition of computed tomography image was undertaken. Radiation treatments were delivered using parallel-opposed ports (6 cm × 6 cm) with 15- to 45-degree wedges, to ensure homogenous dose distribution in the glottic area. Gross tumor volume (GTV) was delineated based on thin-slice (1 mm thickness) CT images, with reference to findings of video fiberscope. Clinical target volume (CTV) encompassed the whole larynx including thyroid cartilage, epiglottis, hyoid bone, and cricoid, regardless of T stage. Planning target volume (PTV) was defined as a 5 mm margin from the CTV. Radiation fields were configured to include hyo-thyro-epiglottic space and level VI lesions, and dose distributions were carefully designed, especially for tumors with ACI. In particular, the dose to the AC was carefully considered and a homogenous dose distribution was planned. Our institution’s policy is to administer conventional radiotherapy (66 Gy/33 Fr, once daily) for T1a disease. However, when the tumor was diagnosed as T1a and bulky, hyperfractionated radiotherapy (twice daily) was initiated. Hyperfractionated radiotherapy was primarily used for the treatment of T1b or T2 disease. However, when the tumor was quite small or when the patient’s condition made twice-daily treatment difficult, conventional radiation (70 Gy/35 Fr) was administered. No elective lymph node irradiation was performed. The median total dose was 66 Gy (range, 60–74.4 Gy) for patients with T1aN0 cancer, 74.4 Gy (range, 70–74.4 Gy) for patients with T1bN0 cancer, and 74.4 Gy (range, 70–74.4 Gy) for patients with T2N0 cancer. Overall, 10 patients (20%) with T1a, 27 patients (71%) with T1b, and 62 patients (94%) with T2 disease were treated with 74.4 Gy of hyperfractionated radiotherapy. The median overall treatment duration was 46 days (range: 40–60 days).

### 2.3. Salvage Surgery

In general, we followed the criteria for partial laryngectomy reported by Biller et al. [30]. Frontolateral laryngectomy consisted of the removal of the frontolateral part of the ala of the thyroid cartilage, including the ipsilateral vocal cord, the AC, and the paraglottic space. In extended frontolateral laryngectomy, the arytenoids are removed, completely or partially, in addition to those described for frontolateral laryngectomy [31]. For follow-up evaluation to detect local recurrence, all patients were carefully evaluated using video laryngoscopy every month for the first year, every 2 months for the next 2 years, and every 3 months thereafter for a total of at least 5 years, by both radiation oncologists and head and neck surgeons. When recurrent tumor was suspected, biopsy was performed for confirmation.

### 2.4. Statistical Analyses

Statistical analyses were performed using Prism version 6.0. The time to the event was calculated from the start date of radiotherapy to the event of interest. For overall survival (OS) rates, the event of interest was identified as death due to any cause. For local recurrence (LC) rates, the event of interest was the histological confirmation of local recurrence based on biopsy results. The Kaplan–Meier method was assessed to plot survival and recurrence curves. Follow-up duration was estimated for surviving patients. The log-rank test was used to assess the differences in local recurrence rates according to various factors. A *p*-value of 0.05 or less was considered statistically significant.

## 3. Results

### 3.1. Survival

The median follow-up was 5.1 years, ranging from 1.2 to 12.6 years. The 5- and 10-year OS rates were 92 and 72%, respectively. The 5- and 10-year cause-specific survival rates were 98 and 97%, respectively (Figure 2A). During the follow-up period, three patients died of laryngeal cancer. One patient with T2 disease with ACI experienced local recurrence after 2.4 years but did not agree to further treatment because of his age and finally died of the disease 5.6 years after the completion of radiotherapy. One patient who had T2 disease with ACI experienced recurrence at the regional lymph nodes after 2.2 years and received chemotherapy for recurrent disease; however, he died within 3.6 years. Another patient with T2 disease without ACI had pulmonary metastases within 7 years, but no treatment was administered because of his health condition and age.

### 3.2. Local Control

The 10-year LC rates were 94% in patients with T1a disease, 88% in patients with T1b disease, and 81% in patients with T2 disease (Figure 2B). Local recurrence occurred in 18 (12%) patients. In the total series of patients, LC with ACI was worse than LC without ACI; however, the difference was not significant (*p* = 0.08; Figure 3A). Subsequently, subanalyses evaluating the existence and size of ACI-affected local controllability were performed by stratifying AC0, AC1, and AC2. The LC rate of patients with AC2 tumors was worse than that of patients with AC0 tumors (*p* = 0.06, Figure 3B). Among patients with ACI, the size of the ACI was not significantly different between AC1 and AC2 (*p* = 0.25, Figure 3C).

The importance of ACI was investigated according to the T stage. LC rates in patients with ACI were slightly better than those in patients without ACI with T1a (*p* = 0.61) or T1b (*p* = 0.38) diseases, but the difference was not significant (Figure 4A,B). In contrast, in patients with T2 disease, the LC rates in patients with ACI were significantly worse than those in patients without ACI (*p* = 0.04, Figure 4C).

### 3.3. Salvage Treatments and Laryngeal Preservation

The 10-year laryngeal preservation rate was 96% (Figure 5). Among 18 (12%) patients who experienced a local-regional recurrence, 15 (9.8%) underwent salvage surgery. Of them, seven patients underwent TLM, four underwent hemilaryngectomy, and four underwent total laryngectomy as the first salvage treatment. Two patients with TLM experienced a second recurrence and underwent a total laryngectomy. Two patients with local recurrence and regional lymph node metastases received systemic chemotherapy. One patient refused salvage surgery due to his age and physical condition.

### 3.4. Morbidity

As shown in Table 2, regarding acute morbidities, six patients (4%) experienced grade 3 mucositis, and four patients (3%) had grade 3 dermatitis. None of the patients had any grade 4 or 5 morbidities. Regarding late morbidities, eleven patients (11%) experienced grade 2 hypothyroidism, whereas none of the patients had morbidities greater than grade 3.

## 4. Discussion

This study illustrates the prognostic significance of ACI in patients with EGSCC treated with standard or hyperfractionated radiotherapy. The hyperfractionated radiotherapy resulted in favorable LC rates for tumors with ACI in T1a and T1b diseases, whereas the regimen was insufficient for tumors with ACI in T2 disease. Moreover, our strategy showed minimal toxicity and excellent laryngeal preservation. This information is beneficial for overcoming the difficulties associated with ACI for EGSCC.

The prognostic factors for LC in early-stage glottic laryngeal cancer have been discussed for several decades. Among these, ACI appears to be one of the most important tumor-related factors. In their review and meta-analysis, Tulli et al. [19] demonstrated that ACI is a negative prognostic factor for T1 glottic tumors, regardless of the treatment method (radiotherapy or TLM). They also performed a subanalysis, evaluating studies that provided information on T1a and T1b separately, and concluded that ACI was a negative prognostic factor. Hendriksma et al. [32], in their review of the literature, demonstrated that 68% of reports showed that ACI did not have a significant impact on LC, whereas 29% showed an impact on EGSCC treated with radiotherapy. We investigated whether tumor size at the AC, observed using fiberoptic laryngoscopy, influenced LC with our treatment strategy. LC in patients with AC tumors was slightly lower than in patients without AC tumors (*p* = 0.08). The LC of larger AC tumors (AC2) was lower than that of smaller AC1 tumors (*p* = 0.25; Figure 3C). In our series of EGSCC cases, the outcomes of T1 disease with ACI were better than those of the same stage without ACI (Figure 4A,B). We speculate that these outcomes were related to the use of hyperfractionated radiotherapy. In contrast, this strategy was insufficient for T2 disease with tumors and ACI in terms of LC (*p* = 0.04; Figure 4C). Hendriksma et al. [32] also demonstrated that no studies in which patients were treated with radiotherapy used a detailed stratification of ACI; therefore, ACI, as well as radiotherapy, may be a risk factor for TLM. They also mentioned that none of the radiotherapy studies showed a 5-year laryngeal preservation rate with respect to ACI. Above all, the data illustrated in this study seem important, in that the existence of ACI and the size of AC tumors seem to be critical factors for LC in ESGCC.

Radiation is an important prognostic factor for treatment. A Japanese trial by Yamazaki et al. [24] randomized 180 patients with T1 disease to receive either 60–66 Gy in 30–33 fractions (2 Gy/fraction) or 56.25–63 Gy in 25–28 fractions (2.25 Gy per Gy/fraction). The 5-year LC rate was 77% and 92%, respectively (*p* = 0.004). They concluded that a higher dose per fraction of 2.25 Gy with a shorter overall treatment time resulted in superior LC. In addition, several retrospective studies have reported the use of 2.25 Gy fraction sizes. A large retrospective series of 585 patients treated at the University of Florida for T1/2 glottic tumors was recently reported, with 61% of the patients receiving ≥2.25 Gy/fraction. A series of 398 patients treated at the University of California was reported in 1997, including a range of fraction sizes; the 5-year LC rate was 85% for T1 and 70% for T2 disease. A fraction size ≥ 2.25 Gy was a significant factor for LC in T2 tumors [33]. They reported a 3–25% decrease in LC rate with a 1-week prolongation of the overall treatment time (OTT). Acceleration of the treatment time can be achieved by either hypofractionation or hyperfractionation with two fractions per day, which is considered to shorten the OTT and improve LC rates. Hyperfractionation is one of the alternative approaches for accelerating treatment schedules. The Radiation Therapy Oncology Group (RTOG) 9512 of a hyperfractionation study of 250 patients with T2N0 glottic carcinoma evaluated between 1996 and 2003 compared a conventional arm of 70 Gy in 35 daily fractions with 79.2 Gy in 66 fractions of 1.2 Gy twice-daily fractions; the 5-year LC rates of the arms were 70% and 78%, respectively (non-significant, *p* = 0.14) [26]. Meta-analyses have shown that accelerated hyperfractionation and hypofractionation improve LC in EGSCCs [34]. Both regimens shortened the OTT compared to standard fractionation, which is consistent with the fact that OTT is directly associated with LC rates. The LC rates in our study were comparable to those reported in previous studies of hypofractionated radiotherapy. Therefore, our strategy of hyperfractionated radiotherapy (74.4 Gy/62 fractions) may be a useful option for EGSCC.

Accelerated hyperfractionation may cause relatively severe acute toxicity such as mucositis [35,36], although several reports have suggested the benefit of accelerated hyperfractionation for EGSCCs [37,38]. In our study, the total dose of hyperfractionated radiotherapy (74.4 Gy/62 fractions) was slightly smaller and resulted in excellent LC and minimum toxicities. Therefore, we plan to continue using this treatment strategy in the future.

The limitations of this analysis were that the results and data were retrospectively obtained from a single institution. Therefore, multi-institutional prospective studies should be conducted in future to verify our findings. Moreover, this study cohort was not evaluated using Magnetic Reso-nance Imaging (MRI) to determine whether invasion to the paraglottic space existed in T2 disease. An undetected invasion into the paraglottic space may be classified as T3 disease. Therefore, for more accurate staging of T2 or T3 disease, MRI or ultrasonography could be beneficial for further evaluation [39,40].

## 5. Conclusions

ACI is an important factor in the management of EGSCC. Regarding LC, hyperfractionated radiotherapy seemed to be effective for T1 disease with ACI, whereas it was insufficient for T2 disease with ACI. However, hyperfractionated radiotherapy contributed to excellent treatment outcomes in terms of laryngeal preservation. To improve local control, more aggressive strategies, such as chemoradiotherapy, may be considered after evaluation with MRI.

## Figures and Tables

**Figure 1 cancers-16-01850-f001:**
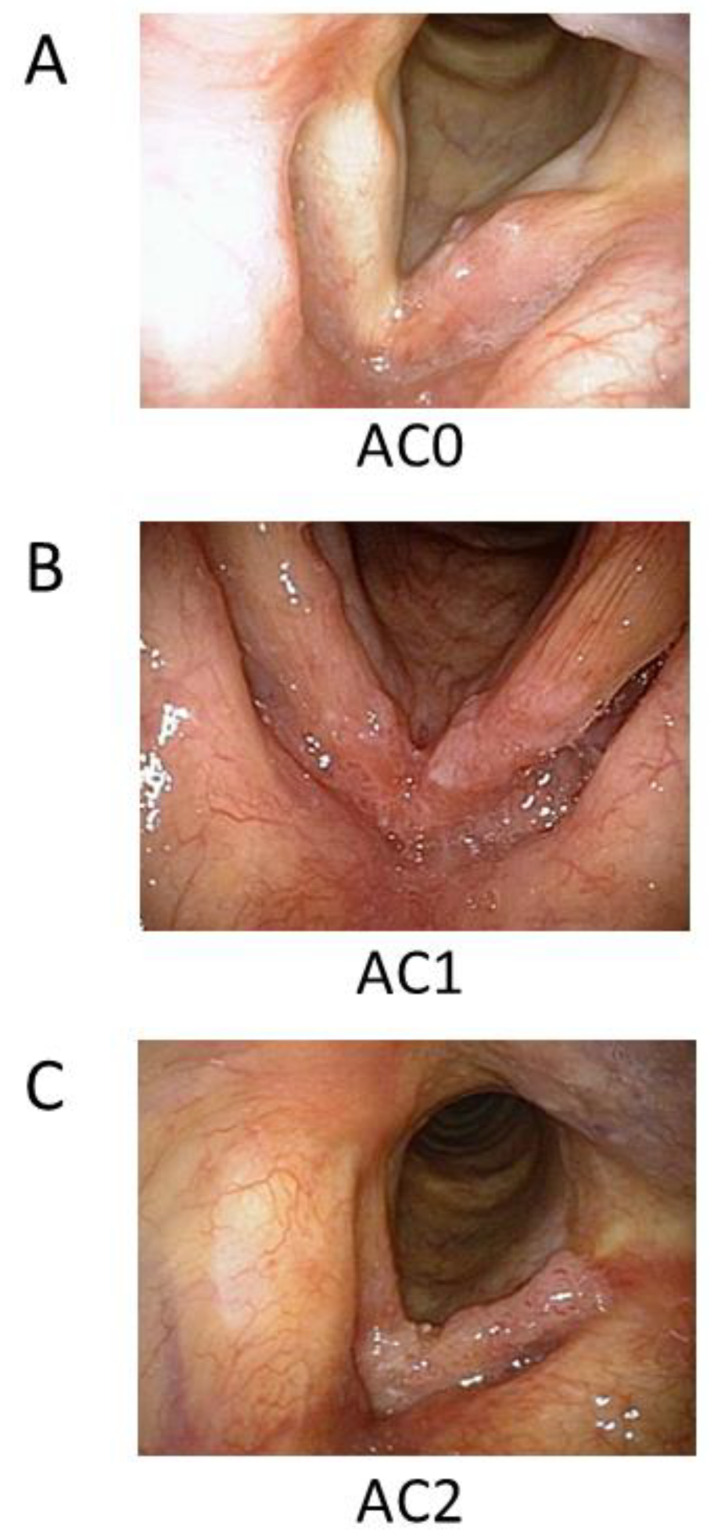
Classification of glottic cancer according to tumor size and ACI. (**A**) AC0 is a tumor without ACI, (**B**) AC1 is a small tumor with ACI, and (**C**) AC2 is a bulky tumor with ACI. ACI, anterior commissure involvement.

**Figure 2 cancers-16-01850-f002:**
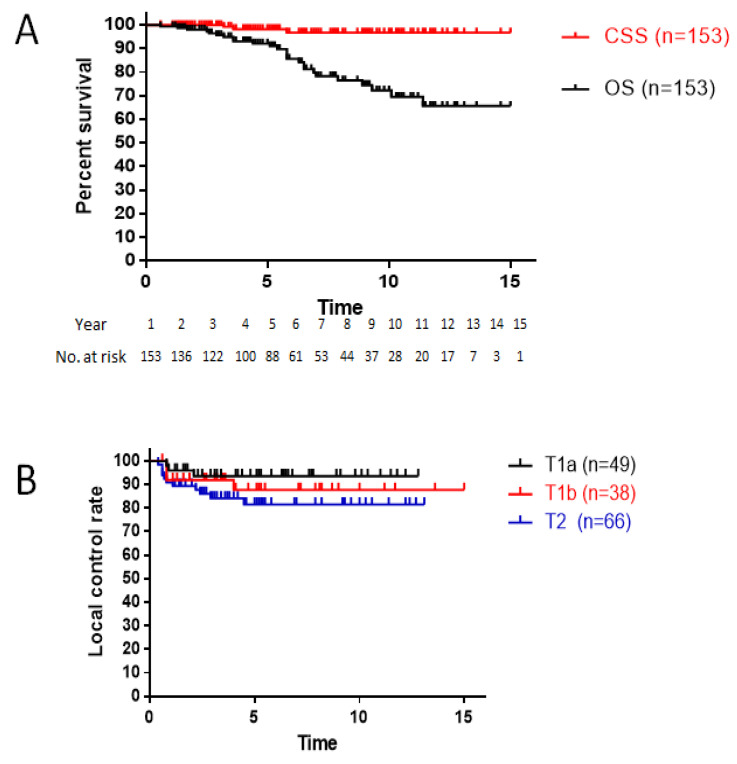
(**A**) Overall and cause-specific survival curves for all patients (n = 153). (**B**) Local control rate according to T stage. CSS, cause-specific survival; OS, overall survival.

**Figure 3 cancers-16-01850-f003:**
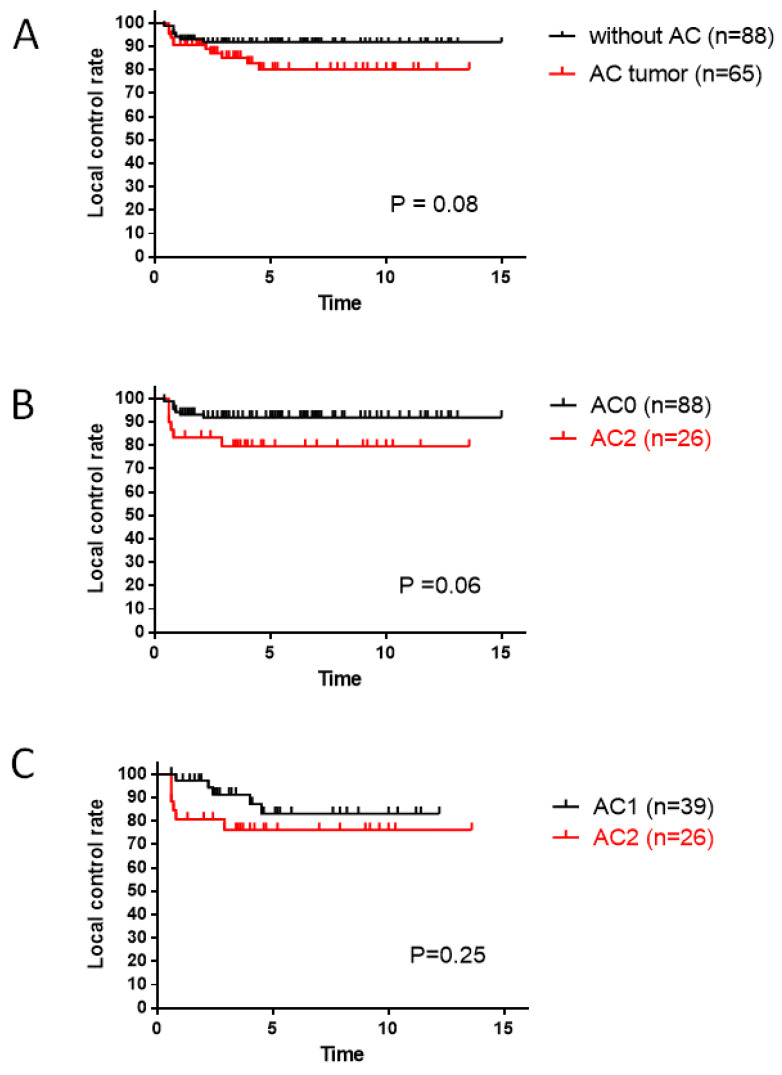
(**A**) Local control rate according to the presence of ACI, (**B**) local control rate according to the presence or absence of ACI, and (**C**) local control rate according to the ACI tumor size classification. ACI, anterior commissure involvement.

**Figure 4 cancers-16-01850-f004:**
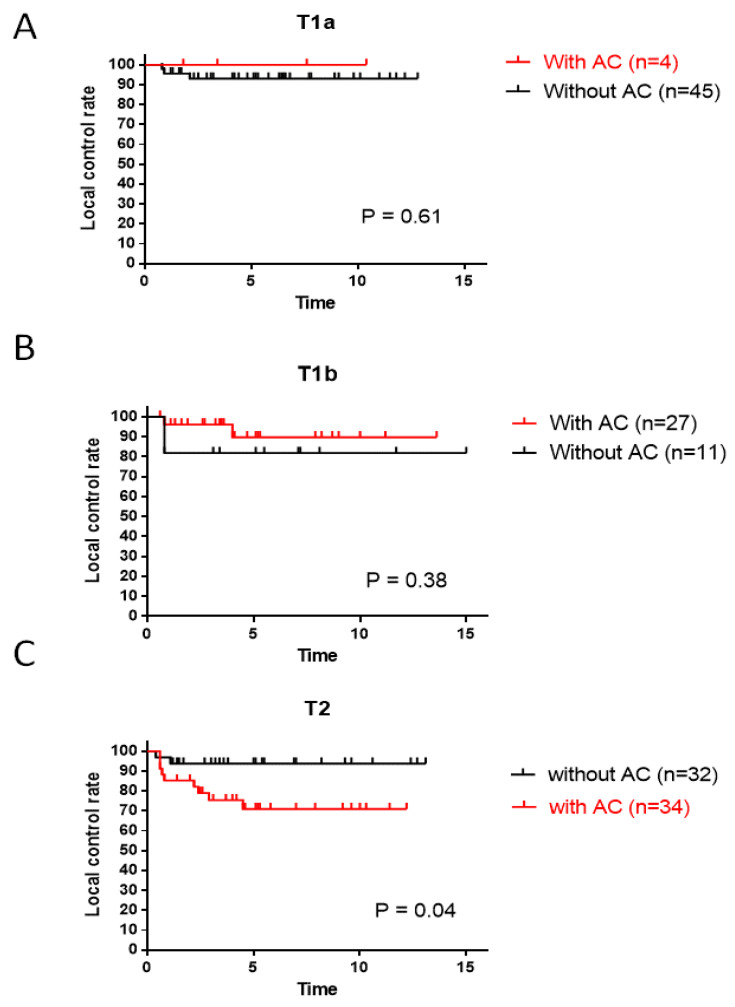
Effect of ACI on the local control of EGSCC. (**A**) T1a, (**B**) T1b, and (**C**) T2 disease. ACI, anterior commissure involvement; EGSCC, early-stage glottic squamous cell carcinoma.

**Figure 5 cancers-16-01850-f005:**
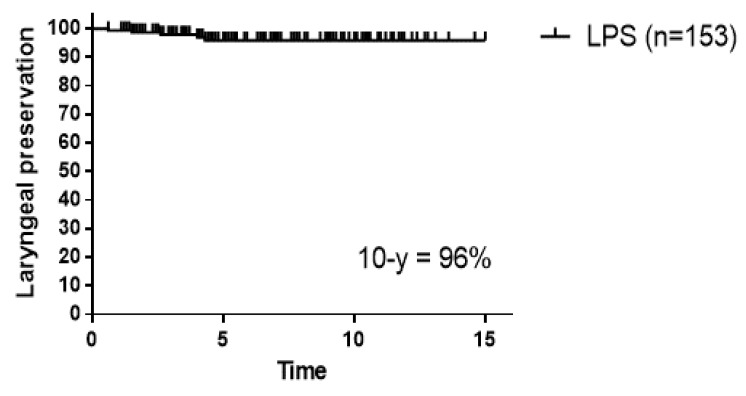
Laryngeal preservation rate in all patients (n = 153).

**Table 1 cancers-16-01850-t001:** Patients’ characteristics and treatments.

		Number (n = 153)	(%)
Age (range)		72 (47–91)	
Gender	Male	136	89
	Female	17	11
ECOG Performance Status	0	63	41
	1	70	46
	2	5	3
	3	1	1
	unknown	14	9
Stage	T1aN0M0	49	32
	T1bN0M0	38	25
	T2N0M0	66	43
Extent of AC involvement	AC0	88	58
	AC1	39	25
	AC2	26	17
Radiation doses, fractions	66 Gy/33 Fr	34	22
	70 Gy/35 Fr	20	13
	74.4 Gy/62 Fr	99	65
Total treatment days Median (days)		48 (range: 43–64)	

ECOG, Eastern Cooperative Oncology Group; AC, anterior commissure.

**Table 2 cancers-16-01850-t002:** Acute and late morbidities associated with radiotherapy.

Grade	0	1	2	3	4	5
Acute						
Mucositis	1 (0.7%)	68 (44%)	78 (51%)	6 (4%)	0	0
Dysphagia	38 (25%)	105 (69%)	10 (7%)	0	0	0
Dermatitis	0	107 (70%)	42 (28%)	4 (3%)	0	0
Late						
Hypothyroidism	127 (83%)	15 (10%)	11 (7%)	0	0	0

## Data Availability

The datasets used and analysed during the current study are available from the corresponding author on reasonable request.

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
