# Peer review of "Targeting Anterior Commissure Involvement with Hyperfractionated Radiotherapy for T1–T2 Squamous Cell Carcinoma of the Glottic Larynx"

_cancers, 2024, doi:10.3390/cancers16101850_

Round 1

Reviewer 1 Report

Comments and Suggestions for Authors

Dear authors,

I read with great interest your manuscript titled "Targeting Anterior Commissure Involvement with Hyperfractionated Radiotherapy for T1–T2 Squamous Cell Carcinoma of the Glottic Larynx". The study presents valuable insights into the treatment outcomes and prognostic factors for EGSCC with ACI. The study is well-conducted and provides important clinical implications for the management of this condition.

The introduction provides the appropriate insights on the disease and the importance of the anterior commissure involvement. However, the aim of the study should be clearly stated, without providing any findings in the introduction section.

In addition, it would be helpful to discuss potential limitations and biases of the study, such as potential confounding variables that were not accounted for in the analysis.

The discussion could also explore the clinical implications of the findings in more detail, including the potential impact on treatment guidelines, patient management strategies, and future research directions.

Reviewer 2 Report

Comments and Suggestions for Authors

Dear Authors,

I read with great interest your article about the use of radiotherapy for the management of early laryngeal carcinoma. However, some aspects require your attention.

In Table 1, please define in the text the variable named Performance status. Performance of what?

You have very few patients that receive chemotherapy concomitant with radiation therapy - only 7 cases. Why did you not exclude these cases from the study?

Moreover, at Radiation doses, you write that you have 1 case that received a personalized radiation fractioning. Why did you not exclude this case from the study?

Furthermore, in Treatment policy, you have 4 subtypes: 66/33 which seems the standard used widely, 70/35 without chemotherapy, 70/35 with chemotherapy (actually these are only the 7 cases mentioned before), and 74.4/62 which is aggressive and the most used in your study. I believe that you could refine the results to exclude the cases with personalized treatment and keep only 2 subgroups 66/33 and 74.4/62.

Your study spans from 2008 to 2021, which means 13 years. Please insert a paragraph mentioning the distribution of cases per year of introduction to the study. Because you have different survival curves. You have according to the Kaplan Meyer curves in the manuscript patients followed for 15 years - January 2008 to January 2024. But other patients have only a 3-year survival rate because you enlisted them in November 2021, you should mention this to the limitations of the study.

From Figure 3 it appears that the cases with AC1 have a lower survival than cases with AC2. How do you explain this? Could this be a result of the radiation regimen used? Maybe the AC2 cases received the more aggressive radiation protocol and therefore the AC1 radiation regimen needs to be upgraded to a more aggressive protocol. 

In the Discussion section, you need to expand on the imaging protocol for diagnosis. The imaging of the tumors of the vocal cords includes sonography of the larynx, CT, and MRI with contrast media. Please reference this to the article by Cergan R, Dumitru M, Vrinceanu D, Neagos A, Jeican II, Ciuluvica RC. Ultrasonography of the larynx: Novel use during the SARS-CoV-2 pandemic (Review). Exp Ther Med. 2021 Mar;21(3):273. doi: 10.3892/etm.2021.9704. Epub 2021 Jan 25. PMID: 33603880; PMCID: PMC7851652.

There are many abbreviations in the text, so I recommend inserting a list of abbreviations at the end of the article.

Before the conclusions, you need to insert a subsection about the limitations of the present study.

Looking forward to receiving the improved version of your manuscript.

Reviewer 3 Report

Comments and Suggestions for Authors

I have read the article by Seno and colleagues with great interest.
The topic is highly relevant to those involved in radiotherapy of the head and neck region.
Early stage laryngeal disease has always been a subject of discussion with ENT colleagues.
Based on my reading of the paper, I would like to offer some considerations.
A total of 161 cases of treatment over a long period of time (2008 to 2021) will be the subject of analysis.
The RT techniques used and how they were planned should be defined and described.
In my opinion, there is a bias in terms of the definition of the disease for the treatment plan.
Authors talk about target definition using thin-slice CT: ask for more definition. CT with mdc, slice thickness? Has MRI never been used?
It may be prudent to exclude very short follow-up from analysing (minimum range follow up, 2 months!).
There seems to be some ambiguity in the definition of disease stage, and the use of MRI to confirm the absence of disease in the paraglottic space should be considered. 
In 7 cases, a combined CRT treatment was given for T2 bulky, which is similar to T3.
This parameter may benefit from a more precise definition.
Additionally, it is unclear why lymph node irradiation was not performed in these cases.
Historicallu, tumors with anterior commissure involvement have been treated with irradiation of the hyo-thyro-epiglottic space and level VI, which should be carefully evaluated.
The current version of the work may be difficult to place and could benefit from further development.
It may be worth considering some revisions to the work to improve its clarity and comprehensibility. Attention to certain details could be beneficial in this regard. 

Comments on the Quality of English Language

English revision should be considered.
